# Sleep Measurement Using Wrist-Worn Accelerometer Data Compared with Polysomnography

**DOI:** 10.3390/s22135041

**Published:** 2022-07-04

**Authors:** John D. Chase, Michael A. Busa, John W. Staudenmayer, John R. Sirard

**Affiliations:** 1Department of Kinesiology, University of Massachusetts Amherst, Amherst, MA 01003, USA; jchase@umass.edu; 2Institute for Applied Life Sciences, University of Massachusetts Amherst, Amherst, MA 01003, USA; mbusa@umass.edu; 3Department of Mathematics & Statistics, University of Massachusetts Amherst, Amherst, MA 01003, USA; jstauden@math.umass.edu

**Keywords:** accelerometer, polysomnography, algorithm, sleep, Cole–Kripke

## Abstract

This study determined if using alternative sleep onset (SO) definitions impacted accelerometer-derived sleep estimates compared with polysomnography (PSG). Nineteen participants (48%F) completed a 48 h visit in a home simulation laboratory. Sleep characteristics were calculated from the second night by PSG and a wrist-worn ActiGraph GT3X+ (AG). Criterion sleep measures included PSG-derived Total Sleep Time (TST), Sleep Onset Latency (SOL), Wake After Sleep Onset (WASO), Sleep Efficiency (SE), and Efficiency Once Asleep (SE_ASLEEP). Analogous variables were derived from temporally aligned AG data using the Cole–Kripke algorithm. For PSG, SO was defined as the first score of ‘sleep’. For AG, SO was defined three ways: 1-, 5-, and 10-consecutive minutes of ‘sleep’. Agreement statistics and linear mixed effects regression models were used to analyze ‘Device’ and ‘Sleep Onset Rule’ main effects and interactions. Sleep–wake agreement and sensitivity for all AG methods were high (89.0–89.5% and 97.2%, respectively); specificity was low (23.6–25.1%). There were no significant interactions or main effects of ‘Sleep Onset Rule’ for any variable. The AG underestimated SOL (19.7 min) and WASO (6.5 min), and overestimated TST (26.2 min), SE (6.5%), and SE_ASLEEP (1.9%). Future research should focus on developing sleep–wake detection algorithms and incorporating biometric signals (e.g., heart rate).

## 1. Introduction

Sleep regulates all major physiological systems and can have a clinically meaningful impact on metabolic and cognitive health [1,2,3,4,5,6,7,8]. The accurate and reliable measurement of sleep metrics is, therefore, essential due to the mediating effect of sleep on numerous health outcomes [1,2,3,4,5,6,7,8]. Polysomnography (PSG) is the gold standard tool for understanding physiological processes related to sleep. However, PSG is often cost-prohibitive, collected in clinical settings unfamiliar to participants, and requires trained technicians to operate and process data. The cumbersome instrumentation used with PSG can shorten sleep duration and decrease sleep quality, particularly during the first night experiencing PSG [9]. Wrist-worn accelerometers provide an inexpensive alternative for the multi-night assessment of several sleep characteristics. Accelerometers can be deployed in ecological settings and provide a less obtrusive method of sleep measurement with the dual capability of measuring other human behaviors, such as physical activity and sedentary behaviors.

Consumer wearable accelerometers continue to grow in popularity, partially due to increased accessibility, social trends, and technological advancement [10]. Since consumers can rely heavily on wearable devices to provide behavioral information, including sleep characteristics, focusing on metric accuracy will allow for improved public health recommendations. As consumer-grade devices do not provide access to raw data, evaluating candidate algorithms from research-grade device data is a necessary step in understanding metric performance before applying the algorithms in a wider context.

Previous studies in healthy adults have detailed the tendency for accelerometers to overestimate sleep and underestimate waking, compared with polysomnography [11,12,13,14]. These systematic errors often occur when accelerometers poorly distinguish between sleep and sedentary supine waking periods (e.g., lying recumbent while reading or watching television) [12,15,16,17]. The Sleep Onset Latency (SOL) period is a primary example of a sedentary supine waking period susceptible to high misclassification error. Varying methods exist for defining SOL, and the method is often dictated by either the algorithm [12,18,19] or user-defined epoch scoring settings [20,21,22]. For instance, the widely used Cole–Kripke algorithm [23] requires 60 s epochs for sleep–wake scoring, while the Oakley algorithm can be applied with 15, 30, 60, or 120 s epochs at low, medium, and high sensitivity count thresholds [20,21,22]. Alongside algorithm selection, researchers can define how sleep onset is scored. For instance, sleep onset may be defined as the first ‘sleep’ epoch, while other operational definitions of sleep onset require multiple consecutive epochs of ‘sleep’ [19,24]. Accordingly, recent efforts have been made to improve the accuracy of accelerometer classification during the SOL period in patients with Obstructive Sleep Apnea and/or Periodic Limb Movement Disorder [15], as well as children and adolescents [25].

Still, few studies have addressed accelerometer errors due to the misclassification of epochs as ‘Sleep’ or ‘Wake’ during periods of the wake-to-sleep transition, particularly during SOL, despite the widespread use of accelerometers in sleep analyses. While optimal sleep onset scoring lengths have been proposed for adults with Obstructive Sleep Apnea and Periodic Limb Movement in Sleep (5 min) [15], as well as children (3 min) and adolescents (20 min) [25], there is no established sleep onset ruling for healthy adults. Sleep onset periods of 1, 5, and 10 min are commonly available settings. The optimization of sleep–wake detection algorithms, particularly during periods of high misclassification, would enhance the utility of accelerometers to provide a more accurate assessment of avital health behavior. Therefore, the purpose of this study was to determine if applying alternative sleep onset (SO) definitions (1, 5, 10 min of immobility), that have been most commonly used in other populations, would improve the accuracy of the commonly used Cole–Kripke algorithm in the classification of accelerometer-derived sleep measurements in healthy adults, compared with PSG.

## 2. Materials and Methods

### 2.1. Participants

Twenty healthy participants (50%F; Age = 24.6 ± 2.7 years; BMI (mean ± SD = 24.3 ± 3.6 kg/m^2^ [Range = 19.4–31.7 kg/m^2^])) completed a 48 h visit within a laboratory that simulated an apartment-style home. This was a secondary data analysis from a study designed to validate home-based health monitoring technologies. Participants had no known sleep disorders. Due to the “first-night-effect” (decreased sleep duration and quality during initial PSG night) [9], only the second night was used in the present analysis. PSG data were recorded for twenty participants, but one participant was excluded due to PSG equipment malfunction; therefore, data from 19 participants were included for these analyses. Informed consent was obtained from all participants prior to study participation. The study protocol was approved by the University of Massachusetts Amherst IRB.

### 2.2. Study Details

#### 2.2.1. Polysomnography

Participants notified the technicians when they were ready to attempt night-time sleep. A standard PSG montage was used, following International 10-20 System guidelines. An Embletta MPR ST+ Proxy (Embla Systems, Natus Neurology, Middleton, WI, USA) was used to sample data at 500 Hz. The MPR was equipped with 16-channel electroencephalography leads (EEG; F3, F4, C3, C4, O1, O2, M1, M2, GRD/REF), electromyography (EMG; 3 sub-mental leads), and electrocardiography (ECG; R/L Arm). Participants also wore thoracic and abdominal respiratory belts, a nasal cannula, and thermistor, all of which were synced to the MPR unit. All impedances were under 10 kΩ upon completion of PSG setup. PSG and accelerometer data were temporally aligned for epoch-by-epoch comparison and merged by participant and timestamp. All devices were initialized on the same computers to ensure clock synchronization.

#### 2.2.2. Accelerometers

Participants wore an ActiGraph GT3X+ (ActiGraph, Pensacola, FL, USA) tri-axial accelerometer (AG) on their non-dominant wrist. The accelerations were sampled at 80 Hz by a 12-bit analog to digital converter with a dynamic range of +/− 6G. Upon study completion, data were downloaded and extracted from the AG non-volatile memory. The accelerations from 80 Hz data were collapsed in 1 s epochs using ActiLife software (v. 6.13.4). The 1 s count data were then collapsed into 60 s epochs using a custom R Script (R Studio Software v. 1.2.5042, RStudio, Boston, MA, USA) as per the requirements for the Cole–Kripke sleep scoring algorithm (‘actigraph.sleepr’). The Cole–Kripke algorithm uses a sliding weighted average window to score 60 s epochs as ‘Sleep’ or ‘Wake’ based upon x-axis (anteroposterior body position; parallel to forearm) count data [26]. The data and custom R script are available upon request.

### 2.3. Data Processing

Night-time sleep was recorded on both nights by an American Academy of Sleep Medicine (AASM)-certified technician. Sleep stages were scored in 30 s epochs by an AASM-certified technician according to AASM standard procedures [27]. Sleep technicians noted ‘Lights Out’ and ‘Awake’ time. Awake was identified as the last PSG epoch scored as ‘Sleep’. Time in Bed (TIB) was identified as the time between ‘Lights Out’ and ‘Awake’.

From the included participants (*n* = 19), portions of PSG data were unable to be scored from *n* = 6 participants (range = 17 to 246 min missing); these missing epochs were excluded from further processing and analyses, in both the PSG and ActiGraph data. The majority of disrupted PSG data instances (*n* = 4) occurred at the end of the night, which simply truncated the data from those participants. When PSG data were interrupted in the middle of the night, the corresponding accelerometer and unscored PSG data were removed. Data from the beginning and end of the night were then temporally synchronized between devices. The first 30 s epoch of every minute of PSG data was taken as the representative value for that minute and time aligned with the corresponding AG minute, to satisfy the Cole–Kripke algorithm requirement (60 s epoch) and eliminate any intra-minute conflicts in the PSG data.

Sleep Onset (SO) was defined by three commonly applied rules based on 1, 5, and 10 consecutive minutes of ‘Sleep’ with no ‘Wake’ score (SO_1-min_ [28,29], SO_5-min_ [15], SO_10-min_ [18,30,31], respectively; Figure 1). The following sleep measures were extrapolated from the PSG data using the three SO rules: Total Sleep Time (TST; total minutes scored as ‘Sleep’ within TIB range), Sleep Onset Latency (SOL; total minutes scored as ‘Wake’ between ‘Lights Out’ and ‘Sleep Onset’), Sleep Efficiency (SE (%) = (TST/TIB) × 100), Efficiency Once Asleep (SE_ASLEEP (%) = TST/(TIB-SOL) × 100), and Wake After Sleep Onset (WASO; total minutes scored as ‘Wake’ between ‘Sleep Onset’ and ‘Awake’). Analogous sleep measures were calculated from AG data.

### 2.4. Statistical Analysis

Agreement between devices, at the 60 s epoch level, was calculated as the percentage of correctly identified ‘Sleep’ and ‘Wake’ epochs by AG, using the PSG scored data as the criteria. Sensitivity was calculated as the percentage of correctly identified ‘Sleep’ epochs. Specificity was calculated as the percentage of correctly identified ‘Wake’ epochs. Cohen’s *d* effect sizes were used to assess differences in sleep measures, according to the device (PSG vs. AG) and SO rule (SO_1-min_, SO_5-min_, and SO_10-min_). To allow comparability with previous studies, standard Cohen’s *d* effect size ranges were used to determine the magnitude of effect (negligible = < 0.2; small = 0.2–0.49; moderate = 0.5–0.79; large ≥ 0.8) [32]. Linear mixed-effects regression models were used to further corroborate the calculated effect sizes. Participant was used as the random effect, while ‘Device’ and ‘Sleep Onset Rule’ were fixed effects. Full and reduced models were compared using a likelihood ratio test to determine if there was an interaction between main effects (‘Device’ × ‘Sleep Onset Rule’). The interactions were not statistically significant at the alpha = 0.05 level. As a result, we used reduced models that only included main effects of ‘Device’ and ‘Sleep Onset Rule’ for all sleep measures.

## 3. Results

The median and interquartile range for each sleep measure is presented in Table 1.

### 3.1. Epoch-by-Epoch Agreement, Sensitivity, and Specificity

Binary epoch-by-epoch sleep/wake scores were used to calculate the agreement, sensitivity, and specificity of AG, compared with PSG (Table 2). ‘Sleep Onset Rule’ did not have a meaningful impact on agreement, sensitivity, or specificity. Sleep/wake agreement and sensitivity were high (89.0–89.5% and 97.2%, respectively), while specificity was low (23.6–25.1%).

### 3.2. Analysis of ‘Sleep Onset Rule’ and ‘Device’

There was no effect of ‘Sleep Onset Rule’ for any sleep measure. According to the effect size analyses (Table 3), the AG significantly underestimated SOL (*d* = 1.09 to 1.46) and WASO (*d* = 0.31 to 0.42), compared with PSG (1 min). Conversely, AG significantly overestimated TST (*d =* −0.28 to −0.31), SE (*d* = −1.18 to −1.37), and SE_ASLEEP (*d* = −0.53 to −0.64).

The linear mixed model estimates for ‘Device’ and ‘Sleep Onset Rule,’ along with their 95% confidence intervals, are represented in Figure 2. There were no significant interactions or main effects of ‘Sleep Onset Rule’ for any sleep measure. Consistent with the effect size analyses, there was a main effect of ‘Device’ where AG overestimated TST (26.2 min), SE (6.5%), and SE_ASLEEP (1.9%), but underestimated SOL (19.7 min) and WASO (6.5 min).

## 4. Discussion

The purpose of this study was to determine if applying alternative sleep onset (SO) definitions, which have been beneficial in improving measurement accuracy in other populations, would improve the accuracy of the commonly used Cole–Kripke algorithm in the classification of accelerometer-derived sleep measurements in healthy adults, compared with PSG. The AG underestimation of SOL was the primary source of the overestimation of TST. However, alternative SO rules (SO_1-min_, SO_5-min_, SO_10-min_) did not impact agreement statistics and there was no main effect of SO rule for any sleep metric.

### 4.1. Agreement, Sensitivity, and Specificity

In the present study, agreement and sensitivity were high, whilst specificity was low. Our results are consistent with most studies demonstrating moderate-to-high agreement and sensitivity between AG and PSG at the expense of specificity [33]. The high agreement may be partially due to the population studied. When Sleep Efficiency is high (>80%), as seen in typical healthy adults [34,35], accelerometers tend to overestimate sleep epochs and underestimate waking epochs [12,36,37]. The current study is consistent with this previous research; agreement between devices was high, and AG-derived TST and SE were both overestimated. These data are congruent with several similar studies on healthy or athletic populations [12,36,37].

### 4.2. Total Sleep Time

The AG overestimated TST, compared to PSG. The tendency for accelerometers to overestimate TST has been well-documented using a variety of devices and algorithms [12,19,28,29,30,36,38,39]. Relatively few studies have reported the underestimation of TST [40,41] or null findings [19,41,42,43]. Regardless of the SO rule applied in the current study, the AG overestimated TST by 26.2 min, which is within the range of previously reported overestimations of TST using the same device and algorithm (8.1 to 81.1 min) [19,28,36].

In the present study, we used the ActiGraph GT3X+ with the Cole–Kripke algorithm. Despite the widespread use of the GT3X+ device to measure patterns of activity and inactivity, few studies have used the same device to estimate sleep measures in healthy adults [19,28,36]. Two of these studies reported relatively large TST overestimations (62.3, 81.1 min) [28,36], compared with the present study (26.2 min). One previous study reported no significant difference in TST between AG and PSG (14.0 min) [19]. Interestingly, the two studies demonstrating large overestimations in TST included middle-aged participants with wide Body Mass Index (BMI) ranges (20–45, 17.7–45.2) [28,36]. The study reporting no significant differences in TST was performed in a healthy young adult population, with a cohort of similar age and anthropometrics to the present study [19]. These data suggest age and health status (e.g., BMI) may impact estimates of sleep metrics, including TST. Together, the ActiGraph GT3X+ accelerometer and Cole–Kripke algorithm appears to be better suited for estimating TST in healthy young adults than middle aged and older adults with high BMI.

### 4.3. Sleep Onset Latency

The AG underestimated SOL by 19.7 min, compared to PSG, which is similar to previous studies on healthy adults, regardless of the device or algorithm used [12,21,28,29,36,37,38,41,42,43,44,45,46,47,48]. Studies using the ActiGraph GT3X+ and Cole–Kripke algorithm have reported underestimations of SOL ranging from 6.4 to 15.1 min [19,28,36,46]. To the best of our knowledge, no study to date has demonstrated an overestimation of SOL in healthy adults.

The underestimation of SOL can be attributed to the poor ‘Wake’ epoch detection capability of accelerometers, supported by the low specificity observed in the present study. The underestimation of SOL and low specificity are well-documented issues when using accelerometers to measure sleep [15,17,33]. Studies on children and sleep-disordered adults have aimed to address the issue of SOL underestimation by applying various rules to define sleep onset. For instance, Chae and colleagues used sleep onset definitions of 4, 5, 6, and 15 min immobility with no more than 1 min interrupted by a ‘wake’ score, with sleep-disordered participants (Obstructive Sleep Apnea, Periodic Limb Movement Disorder) [15]. The 5 min rule outperformed other sleep onset definitions [15]. We adopted an approach similar to studies that demonstrated improvements in SOL estimation by applying three Sleep Onset Rules (1 min, 5 min, and 10 min) [15,25]. We did not observe differences for any sleep measures among sleep onset rules, suggesting the need for alternative methods to address SOL underestimation errors due to poor sleep–wake detection.

### 4.4. Wake after Sleep Onset

Our results are similar to previous studies demonstrating an underestimation of WASO, with biases ranging from 1.4 to 60.2 min [12,19,21,28,29,31,36,37,43,46,49,50]. Studies that used the GT3X+ and Cole–Kripke algorithm reported some of the lowest (1.4 min) [19] and highest (46.8–60.2 min) [28,36] underestimations of WASO. The source of WASO underestimation is likely the same as SOL underestimation, which is the poor ‘Wake’ epoch detection of accelerometers. Supporting this claim, the only other study to report specificity while using the GT3X+ and Cole–Kripke algorithm found a low specificity (35%) [19]. While the authors did not report SOL, there was a slight underestimation of WASO (1.4 min) within their sub-sample of healthy young adult men [19].

### 4.5. Sleep Efficiency

In the present study, we observed an overestimation of AG-derived SE, compared with PSG (1.9%), which can be attributed to the inaccurate prediction of sleep during still waking periods (underestimating SOL and WASO). Our results are consistent with other studies using the ActiGraph GT3X+ and Cole–Kripke algorithm, with overestimations of 1.3% [19], 12.6% [28], and 18.3% [36], respectively.

As an exploratory aim, we investigated the influence of removing waking epochs, from the PSG- and AG-derived data, at the beginning of the night, to determine if SE estimates would improve. The effect size for SE_ASLEEP was attenuated, compared with SE, and the regression analysis indicated that SE_ASLEEP was not significantly different from PSG. With the elimination of the wake-to-sleep transition time (SOL; a time that is known to be a source of misclassification), the underestimation of WASO by the ActiGraph improved the SE_ASLEEP accuracy. To the best of our knowledge, no other studies to date have explored the elimination of the wake-to-sleep transition when comparing AG versus PSG-derived SE. The consistent overestimation of SE with the GT3X+ and Cole–Kripke algorithm is likely due to the overestimation of TST, since two of the studies using this device and algorithm also reported the overestimation of TST [28,36]. Supporting this notion, the study reporting the highest overestimation of TST was the same to report the highest overestimation of SE [36].

Although the overestimation of ‘sleep’ epochs (e.g., TST and SE) adversely affect the agreement between AG and PSG, the primary source of error is the underestimation of ‘wake’ epochs (e.g., WASO and SOL). Previous studies reporting the overestimation of ‘sleep’ epochs and the underestimation of ‘wake’ epochs also report disproportionally high sensitivities (96.7–99.0%) and low specificities (14.3–48%) [12,29,37,44,46]. Importantly, these previous studies utilized a variety of devices, algorithms, and user-defined settings, highlighting the universal importance of improving ‘wake’ epoch detection to improve wearable device accuracy.

## 5. Summary and Conclusions

The present study contributes at least two novel additions to the study of accelerometer-derived sleep measures: (1) the exploration of different sleep onset rules to improve sleep–wake detection, a period of the wake-to-sleep transition with previously identified issues of low specificity; and (2) the use of healthy young adults. Previous studies that have explored the use of alternative sleep onset rules have only been conducted on children, adolescents, and sleep-disordered adults [15,51]. Furthermore, the present study accounted for the PSG ‘first-night-effect’ [9] by collecting data for two nights, but only analyzing data from the second night. The utilization of a linear mixed-effects regression represents a methodological advance, allowing us to appropriately address the non-normality in the data and provide directly interpretable metrics of change to the sleep-onset rules.

The primary limitation of the present study was the use of a single algorithm and device with one-wear location. This decision was made to simplify the aims of the present study. Future studies may consider comparing multiple algorithms across several devices and wear locations, which would require a larger sample size for adequate statistical power. Within this narrow scope, our findings indicate that, regardless of sleep onset rule, the Cole–Kripke algorithm applied to AG data misclassifies ‘wake’ periods while a person is lying in bed as ‘sleep’.

More sophisticated approaches, including machine learning techniques, are currently being developed [52,53] as alternative methods that could be used for improving the wake-to-sleep transitions in healthy and diseased populations. Additionally, incorporating biometric signals (e.g., heart rate, respirometry) related to physiological changes that occur as one falls asleep, and are indicative of sleep, may be combined with the same actigraphy-based device to increase the specificity of sleep measures. The development of validated accelerometer data processing methods to improve sleep/wake detection will enhance the portability and accessibility of ambulatory sleep monitoring in clinical populations. Improvements in accelerometer sleep/wake scoring, when developed in conjunction with devices that incorporate biometric signals, will assist in minimizing the financial and physical burdens of laboratory-based polysomnography tests. Improving sleep/wake detection, and doing so in free-living environments, will allow researchers and clinicians to scale-up sleep and health intervention efforts.

## Figures and Tables

**Figure 1 sensors-22-05041-f001:**
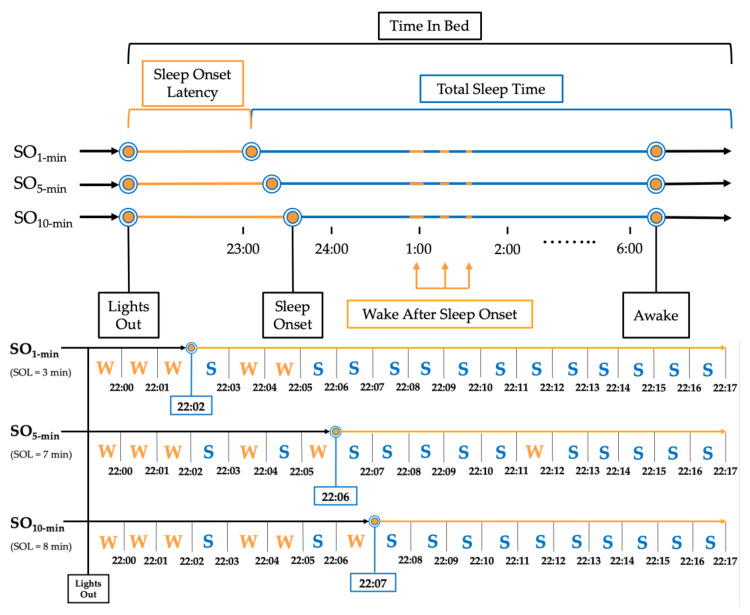
Example of implementation of sleep onset rules for a single participant. Note: Representation of a single night processed using the three sleep onset rules for a mock participant. SO_1 min_ = Sleep Onset defined as first epoch scored as ‘Sleep’; SO_5 min_ = Sleep Onset defined as first 5 consecutive epochs scored as ‘Sleep’; SO_10 min_ = Sleep Onset defined as first 10 epochs scored as ‘Sleep.’.

**Figure 2 sensors-22-05041-f002:**
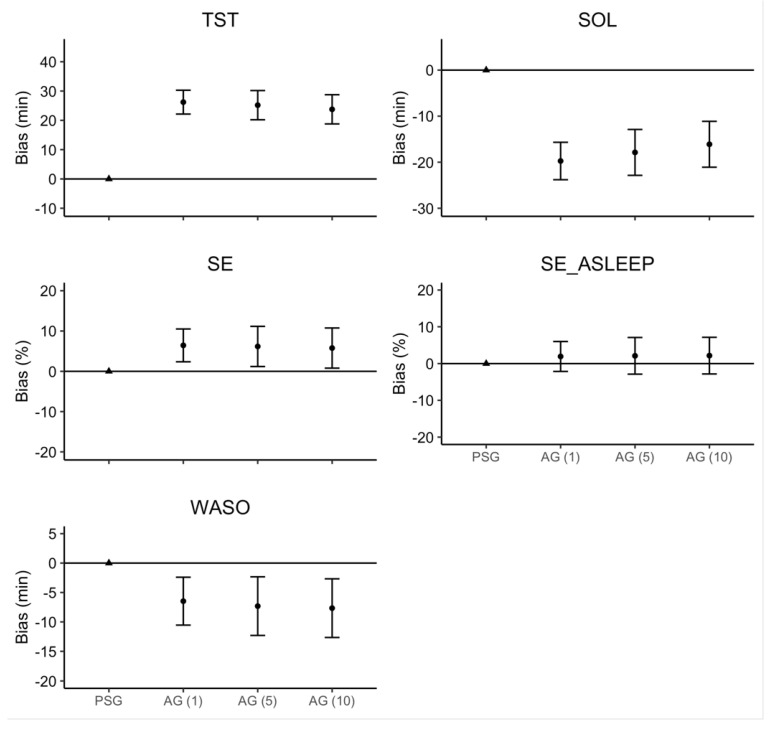
Regression model bias estimates for AG–derived sleep measure. Note: Error bars represent the 95% confidence interval around the bias for each estimate.

**Table 1 sensors-22-05041-t001:** Descriptive statistics of sleep measures for device and sleep onset rule.

	Device Median (IQR)		Sleep Onset Rule Median (IQR)		
Sleep Measure	PSG	AG	1	5	10
**TST** **(min)**	411.0 [321.0–447.0]	431.0 [348.0–465.0]	427.5 [347.3–456.3]	426.0 [347.0–456.3]	426.0 [332.8–453.3]
**SOL** **(min)**	19.0 [10.0–34.0]	1.0 [0.0–8.0]	7.5 [0.3–18.3]	9.5 [1.3–18.3]	11.0 [5.3–23.3]
**WASO** **(min)**	25.0 [13.0–32.0]	16.0 [10.0–29.0]	23.5 [13.5–31.3]	20.5 [13.0–31.3]	20.5 [13.0–31.3]
**SE** **(%)**	89.0 [92.1–96.0]	94.7 [91.5–97.2]	91.7 [89.0–95.5]	91.7 [88.9–95.2]	91.5 [87.6–94.9]
**SE_ASLEEP** **(%)**	93.5 [92.6–96.1]	96.1 [93.2–97.2]	95.5 [93.1–97.0]	96.5 [93.2–97.0]	96.5 [93.4–97.0]

**Table 2 sensors-22-05041-t002:** Epoch-by-epoch agreement, sensitivity, and specificity.

	Agreement (%)	Sensitivity (%)	Specificity (%)
**1**	89.0	97.2	25.1
**5**	89.2	97.2	23.7
**10**	89.5	97.2	23.6

**Table 3 sensors-22-05041-t003:** Effect of sleep onset rule on sleep measures.

	Effect Size Cohen’s *d* (95% CI)				
Sleep Measure	PSG		AG		
	5	10	1	5	10
**TST** **(min)**	0.03 (−0.64, 0.69)	0.01 (−0.68, 0.69)	−0.31 * (−1.03, 0.37)	−0.30 * (−0.94, 0.37)	−0.28 * (−1.01, 0.4)
**SOL** **(min)**	−0.07 (−0.75, 0.60)	−0.13 (−0.78, 0.55)	1.46 ^‡^ (1.11, 2.59)	1.28 ^‡^ (0.92, 2.18)	1.09 ^‡^ (0.72, 1.84)
**WASO** **(min)**	0.02 (−0.64, 0.69)	0.02 (−0.64, 0.74)	0.31 * (−0.40, 0.93)	0.38 * (−0.26, 1.02)	0.42 * (−0.24, 1.00)
**SE** **(%)**	0.05 (−0.64, 0.69)	0.11 (−0.52, 0.77)	−1.37 ^‡^ (−2.28, −0.82)	−1.32 ^‡^ (−2.15, −0.74)	−1.18 ^‡^ (−1.19, −0.60)
**SE_ASLEEP (%)**	−0.02 (−0.70, 0.64)	−0.03 (−0.65, 0.62)	−0.53 ^†^ (−1.22, 0.14)	−0.61 ^†^ (−1.29, 0.00)	−0.64 ^†^ (−1.37, −0.04)

PSG with 1 min sleep onset rule was used as the criterion measure for all effect size comparisons. * = small effect; **^†^** = moderate effect; **^‡^** = large effect.

## Data Availability

Not applicable.

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
