# Peer review of "Sleep Measurement Using Wrist-Worn Accelerometer Data Compared with Polysomnography"

_sensors, 2022, doi:10.3390/s22135041_

Round 1

Reviewer 1 Report

The Discussion section still has the generic prompt included. "Authors should discuss the results and how they can be interpreted from the 196 perspective of previous studies and of the working hypotheses. The findings and their 197 implications should be discussed in the broadest context possible. Future research 198 directions may also be highlighted" Please delete this.

Reviewer 2 Report

In this study, authors investigated sleep measurement with wrist-worn accelerometers. Study is interesting, however, some aspects should be addressed before potential publication:

- Better explanation of “sedentary supine waking periods” needed

- Accompanying text that goes with Results could be written better, with better emphasis on main results

- Lines 196-199 should be deleted, as it is explanation of Discussion writing

- How was sample size needed for this study estimated? Better explanation needed

- In Discussion section, potential clinical implications that can be suggested from the results of this study should be explored

Round 2

Reviewer 2 Report

The authors have answered each of the comments adequately and improved the manuscript. I have no further comments.